# A Pilot Study of Exosome Proteomic Profiling Reveals Dysregulated Metabolic Pathways in Endometrial Cancer

**DOI:** 10.3390/biomedicines13010095

**Published:** 2025-01-03

**Authors:** Feras Kharrat, Valeria Capaci, Andrea Conti, Valentina Golino, Pietro Campiglia, Nour Balasan, Michelangelo Aloisio, Danilo Licastro, Lorenzo Monasta, Federica Caponneto, Antonio Paolo Beltrami, Federico Romano, Giovanni di Lorenzo, Giuseppe Ricci, Blendi Ura

**Affiliations:** 1Institute for Maternal and Child Health, IRCCS Burlo Garofolo, 65/1 Via dell’Istria, 34137 Trieste, Italy; feras.kharrat@burlo.trieste.it (F.K.); valeria.capaci@burlo.trieste.it (V.C.); andrea.conti@burlo.trieste.it (A.C.); nour.balasan@burlo.trieste.it (N.B.); federico.romano@burlo.trieste.it (F.R.); giuseppe.ricci@burlo.trieste.it (G.R.); blendi.ura@burlo.trieste.it (B.U.); 2Department of Pharmacy, University of Salerno, Via Giovanni Paolo II 132, 84084 Salerno, Italy; valentina.golino@unina.it (V.G.); pcampiglia@unisa.it (P.C.); 3National PhD Program in RNA Therapeutics and Gene Therapy, University of Naples Federico II, 80131 Napoli, Italy; 4Functional Gastrointestinal Disorders Research Group, National Institute of Gastroenterology-IRCCS “Saverio de Bellis”, 70013 Castellana Grotte, Italy; michelangelo.aloisio@irccsdebellis.it; 5AREA Science Park, 34149 Trieste, Italy; licastrod@gmail.com; 6Department of Medicine, University of Udine, 33100 Udine, Italy; federica.caponnetto@uniud.it (F.C.); antonio.beltrami@uniud.it (A.P.B.); 7Azienda Sanitaria Universitaria Friuli Centrale, 33100 Udine, Italy; 8Department of Medicine, Surgery and Health Sciences, University of Trieste, 34149 Trieste, Italy

**Keywords:** proteomics, mass spectrometry, exosomes

## Abstract

**Background/Objectives**: Endometrial cancer (EC) is the second most frequent gynecological malignant tumor in postmenopausal women. Pathogenic mechanisms related to the onset and development of the disease are still unknown. To identify dysregulated proteins associated with EC we exploited a combined in vitro/in silico approach analyzing the proteome of exosomes with advanced MS techniques and annotating their results by using Chymeris1 AI tools. **Methods**: To this aim in this pilot study, we performed a deep proteomics analysis with high resolution MS (HRMS), advanced computational tools and western blotting for proteomics data validation. **Results**: That allowed us to identify 3628 proteins in serum albumin-depleted exosomes from 10 patients with EC compared to 10 healthy controls. This is the largest number of proteins identified in EC serum EVs. After quantification and statistical analysis, we identified 373 significantly (*p* < 0.05) dysregulated proteins involved in neutrophil and platelet degranulation pathways. A more detailed bioinformatics analysis revealed 61 dysregulated enzymes related to metabolic and catabolic pathways linked to tumor invasion. Through this analysis, we identified 49 metabolic and catabolic pathways related to tumor growth. **Conclusions**: Altogether, data shed light on the metabolic pathways involved in tumors. This is very important for understanding the metabolism of EC and for the development of new therapies.

## 1. Introduction

Endometrial cancer (EC) is the second most frequent gynecological malignant tumor in the world. The number of new cancer cases increased by 2.3 percent compared to 2020, reaching 2.74 million in 2022 [1,2]. Two types of EC are known: estrogen-dependent type I and estrogen-independent type II. Although type II is less frequent, with an estimated rate of 10% of all EC cases, it is more aggressive and associated with higher rates of recurrence [3].

Extracellular vesicles (EVs) are nanosized vesicles that differ in size, cargo, and markers on the surface, with exosomes being the most studied and characterized. EVs are released into the extracellular environment by different mechanisms and are found in almost all body fluids. EVs play an essential role in cell communication by fusion, phagocytosis, ligand-receptor interaction, and proteolytic cleavage [4]. Moreover, they were reported to be implicated in endometrial pathogeneses and biomarker identification, including EC by immunological impairment in the tumor microenvironment [5]. EVs derived from the female reproductive system have a high prognostic value and may serve as biomarkers for diseases, including EC [6]. Although the proteome of extracellular vesicle proteins in endometrial cancer has been explored by us and others [7,8], most previous work is in cell lines or focuses on specific proteins. Recently, in a study by Song et al., it was reported that the level of Galectin-3 binding protein (LGALS3BP), a glycoprotein found in nearly all body fluids, was elevated in plasma exosomes in EC patients and this increase was associated with the presence of metastases, which suggests this protein as a biomarker for EC [9]. However, an integrated analysis that points to the identification of broad altered pathways is still missing.

Several metabolic alterations occur in cells when they become malignant. The high demand for energy to proliferate and divide necessitates sustainable activation for many pathways related to energy production. Enzymes are the key players in this metabolic reprogramming. For example, most cancer cells rely on aerobic glycolysis, known as the Warburg effect, which was defined as the preference of cancer cells to use glycolysis instead of mitochondrial oxidative phosphorylation (OXPHOS) for energy supply even in the presence of oxygen to produce ATP more quickly [10,11]. Importantly, reprogrammed energy metabolism in the cancer cell alters several pivotal functions, like apoptosis and autophagy, leading ultimately to drug resistance and reduced therapeutic response [12].

Endometrial cancer is related to metabolic abnormalities, and several metabolic pathological conditions are considered risk factors for EC, like obesity, weight change, and metabolic syndrome [13,14].

The proteomic approach is becoming the key approach in cancer pathophysiology studies and for the identification of new biomarkers [15]. In addition, the enhancement of data processing tools for mass spectrometry (MS) with artificial intelligence (AI) has allowed for an increase in the number of identified and quantified proteins [16,17].

In this work, we integrate cutting-edge techniques (Data-Dependent Acquisition (DDA) and Data-Independent Acquisition (DIA) acquisition, and AI) in albumin-depleted serum exosomes to advance the understanding of EC’s molecular and metabolic basis. With this approach, we were able to broaden the landscape of exosomal proteins and enzymes altered in EC patients and, with more proteins, perform more reliable pathway analysis.

## 2. Materials and Methods

### 2.1. Patients

During 2023, a total of 40 patients (20 women suffering from EC and 20 non-EC controls) were recruited at the Institute for Maternal and Child Health—IRCCS “Burlo Garofolo” (Trieste, Italy). Our Institute’s Technical and Scientific Committee approved the study, and all procedures complied with the Declaration of Helsinki. All patients signed informed consent forms. The median age of patients (Appendix A) was 71 years (Min = 42, Max = 84). The median age of controls was 33 years (Min = 25, Max = 61). A physical examination and ultrasound were performed to screen patients for gynecological pathologies. Blood samples were taken after gynecological examination and before the surgery. The inclusion criteria include patients over 18 years of age for both groups, patients with endometrial carcinoma (endometrioid adenocarcinoma) with post-hysteroscopic histological diagnosis (stage FIGO I) for the patients’ group, and patients with non-endometrial pathology with negative endometrial histology as a control group. The exclusion criteria for both groups include patients with infectious diseases (HIV, HBV, HCV, etc.), leiomyoma, and adenomyosis; presence of synchronous tumors; and patients suffering from other neoplastic pathologies and/or being subjected to chemo and radiotherapy in the last 10 years.

### 2.2. Serum Sample Collection and Exosomes Isolation

Blood was centrifuged at 5000 RCF for 5 min to obtain the serum and was collected and stored at −80 °C. The isolation of exosomes from serum with albumin depletion was conducted as previously described [7]. Shortly, 150 µL of crude serum was incubated for 5 min in immobilized Cibacron Blue 3G resin from the Albumin Depletion Kit (Thermo Fisher, Waltham, MA, USA). After column elution, 150 µL of depleted serum was mixed with 20 µL isolation reagent (Total Exosome Isolation Reagent (from serum) and incubated for 30 min at 4 °C. Finally, the samples were centrifuged at 10,000 RCF for 10 min and resuspended with 30 µL of PBS. Protein content was determined using Bradford reagent.

### 2.3. Nanoparticle Tracking Analysis

Analysis of particle size distribution and concentration of purified Exosomes derived from EC patients and controls were performed by Nanosight (LM10, Malvern System Ltd., Malvern, UK), equipped with a 405 nm laser. To perform the particle analysis, each sample was diluted 1:2000 in PBS buffer and recorded for 60 s with a detection threshold set at the maximum. The acquisitions for each sample were made in triple replication.

### 2.4. Western Blotting

Western blotting methodology was conducted as described by [18]. In total, 30 µg of protein from patients and controls were loaded onto 4–20% precast gel and then transferred to a nitrocellulose membrane. After protein transfer, the membrane was blocked by treatment with 5% defatted milk in TBS-tween 20 and incubated overnight at 4 °C with antibodies against CD9 (1:800, rabbit polyclonal), CD64 (1:1000, rabbit polyclonal), HSC70 (1:800, rabbit polyclonal), PRDX2 (1:800, rabbit polyclonal), GADPH (1:800, mouse monoclonal). To remove the excessive presence of the primary antibody, nitrocellulose membranes were washed three times with TBS-Tween 0.05% and incubated with HRP-conjugated anti-rabbit and anti-mouse IgG (1:3000). Primary and secondary antibodies were purchased from Sigma-Aldrich (Merck KGaA, Darmstadt, Germany). Protein band signal visualization was performed by using SuperSignal West Pico Chemiluminescent (Thermo Fisher Scientific Inc., Ottawa, ON, Canada). The intensities of the immunostained bands were normalized with the total protein intensities measured by staining the membranes from the same blot with a Red Ponceau solution (Sigma-Aldrich, St. Louis, MO, USA).

### 2.5. Proteome Analysis

An amount of 30 µg of depleted exosomes was digested with the EasyPep™ MS Sample Prep Kits (Thermo Fisher). Analysis was performed by using nanoflow ultra-high performance liquid chromatography–high-resolution MS using an Ultimate 3000 nanoLC (Thermo Fisher Scientific, Bremen, Germany) coupled to an Orbitrap Lumos tribrid mass spectrometer (Thermo Fisher Scientific) with a nanoelectrospray ion source (Thermo Fisher Scientific) like previously described [19]. One μL of the digest was loaded and trapped on a PepMap trap column for 1.00 min at a flow rate of 40 μL/min (Thermo Fisher), and then peptides were separated by a C18 reversed-phase column (250 mm × 75 μm I.D, 2.0 µm, 100 Å, EasySpray, Thermo). A linear 90 min gradient was performed. MS analysis was performed in DDA, with an MS1 range of 375–1500 m/z set at 120,000 resolution while MS2 was at 15,000. The isolation window was set to 1.6 Da normalized, and a collision energy (HCD) value of 30 was applied.

For MS/MS, the maximum ion injection time for the MS/MS (OT) scan was set to 50 ms, and ACG values were set to standard. The dynamic exclusion was set to 30 s.

For DIA, a first MS scan was performed at 120,000 resolution. DIA was performed in OT at 15,000 resolution. DIA was performed with 10 Dalton isolation windows, and AGC target and maximum ion injection time was set to custom using 200 ms and 40 ms values. Loop control was set to N = 30. HCD was used with a collision energy value of 30.

Proteome Discoverer 3.1 was used for the DDA raw data analyzed using Sequest HT, AMANDA3.0, Chimerys, and INFERYS search engines. The following parameters were used: enzyme trypsin missed cleavages max 2, precursor mass tolerance 10 ppm, and fragment mass tolerance 0.02 Da. Carbamidomethylation was used as a fixed modification, while methionine oxidation was used as the variable. Proteins were considered identified with at least one unique peptide, setting a false discovery rate (FDR) threshold of <1%. The DIA raw data were analyzed using Spectronaut 19. The direct DIA (deep) tool was used for identification. Carbamidomethylation was used for fixed modification, while acetylation and oxidation were used for variable modifications. The FDR < 1% and proteins were considered to be identified with at least one unique peptide. The mass spectrometry proteomics data have been deposited to the ProteomeXchange Consortium via the PRIDE partner repository with the dataset identifier PXD058193.

### 2.6. Bioinformatic Analysis

For bioinformatic analysis, the gProfiler tool (https://biit.cs.ut.ee/gprofiler/gost, accessed on 28 December 2024) was used for exosomes protein characterization, according to their molecular function, protein class, and cellular component. PANTHER was used for protein class, while the Reactome tool (https://reactome.org/PathwayBrowser/#TOOL=AT, accessed on 28 December 2024) was used for pathway identification. The Venn diagram tool (https://bioinformatics.psb.ugent.be/webtools/Venn/, accessed on 28 December 2024) was used for data correlation in proteomics, while SRplot [20] and Morpheus tools (https://software.broadinstitute.org/morpheus/, accessed on 28 December 2024) were used for enriched bar plot and heat maps. The bio-functions were generated via IPA with a significance of *p* < 0.01, as previously described [21].

### 2.7. Statistical Analysis

Differences between patients and controls were considered significant when proteins showed a fold change ≥1.5 and ≤0.6 and satisfied the Mann–Whitney U test (*p* < 0.05). Analyses were conducted using the RStudio script (https://rstudio-education.github.io/hopr/starting.html, accessed on 28 December 2024)

## 3. Results

### 3.1. Exosomal Characterization

The first step for exosome characterization was performing biophysical and biological characterization. We used Nanoparticle tracking analysis (NTA) methodology to measure the concentration and dimension of exosomes from two controls and two EC patients. NTA measurements of CTRL 1 (Figure 1A) showed a vesicle concentration of 1.79 × 10^9^ particle/mL and size distribution with a modal value of 118.4 nm.

In contrast, CTRL 2 showed a vesicle concentration of 8.90 × 10^8^ particles/mL with a modal value of 124.1 nm. EC patient 1 showed a vesicle concentration of 1.09 × 10^9^ particle/mL and a modal value of 141.4 nm. In comparison, EC patient 2 showed a vesicle concentration of 1.59 × 10^9^ particles/mL and a distribution of 124.1 nm. We performed western blotting (Figure 1B) of common exosome markers CD63 and CD9 and cytosolic proteins in EV markers HSC70 to check the good performance of the EV isolation kit (Appendix A). All in one, these data indicate that the vesicles obtained are exosomes.

### 3.2. Proteomics Study

We conducted a proteomics study to explore the protein content of exosomes on 20 samples (10 EC samples and 10 CTRL samples). For deep proteomics analysis, we performed DDA and DIA acquisition. For the DDA data elaboration, we used Proteom Discoverer 3.1. As an initial step, we analyzed the DDA file with different search engines such as: Chymeris, Amanda, Sequest, and Inferis. We combined these tools in order to identify the highest number of proteins (Chimerys, Chimerys + Amanda + Sequest HT, Chimerys + Amanda, Amanda + Sequest HT + Inferis). We identified 472 proteins with Chimerys, 475 with Chimerys + Amanda, 455 with Chimerys + Amanda + Sequest, 352 with Sequest + Amanda + Inferis (Appendix A) with q-value < 0.05 and FDR < 1%. Only Chymeris (34 proteins) and the combination of Sequest + Amanda + Inferis (14 proteins) identified non-common proteins. Finally, by combining the different tools, we identified 515 proteins (Figure 2) (Supplement Appendix A).

The second step was processing Dia files. We used Spectronaut 19 software to process DIA files and identified 3406 proteins with q < 0.01 and FDR < 1%. (Supplement Appendix A). The last step was matching data between the two acquisitions (DDA + DIA) identifying 3628 unique proteins (Figure 3) (Supplement Appendix A).

To the 3628 proteins previously identified, we applied the Mann–Whitney sum-rank test (*p* < 0.05) and the fold change ≥1.5 and ≤0.6. We identified 373 proteins that revealed a significant alteration according to the Mann–Whitney sum-rank test (*p* < 0.05) (Appendix A). Our data proved that 299 were increased in EC with a fold change ≥ 1.5, and 74 proteins decreased in EC with a fold change ≤ 0.6 (Supplement Appendix A).

### 3.3. Bioinformatic Analysis

In the next step, we performed a bioinformatic analysis using the gProfiler tool to understand the functional impact of differently abundant proteins between EC serum exosomes versus serum exosomes from controls (Figure 4).

The bioinformatics tool was used for protein characterization based on their molecular function, biological processes, and cellular components. In terms of molecular function, proteins were ranked into glycosaminoglycan binding, antioxidant activity, catalytic activity, and phosphatidylcholine-sterol O-acyltransferase activator activity. In terms of biological processes, they were ranked by blood coagulation, protein-lipid complex remodeling, regulation of multicellular organism processes, and organonitrogen compound metabolic processes. In terms of cellular component extracellular space, they were ranked by collagen-containing extracellular matrix, endocytic vesicle lumen, and proteasome complex. The Reactome tool for the pathway database indicates that these proteins are primarily involved in neutrophil degranulation, response to elevated platelet cytosolic Ca^2+^, platelet degranulation, and post-translational protein phosphorylation (Figure 5).

Ingenuity Pathway Analysis (IPA) showed that 66 identified proteins are related to the top network corresponding to invasive cancer (Figure 6).

### 3.4. Enzyme Component in Exosomes

We used the PANTHER tool (protein class) to explore the enzyme component in exosomes. We identified 61 dysregulated enzymes (fold change ≥ 1.5 or ≤0.6) (Mann–Whitney *p*-value < 0.05) (Figure 7).

Of them, 49 were more abundant in EC with a fold change ≥ 1.5, and 12 were less abundant with a fold change ≤ 0.6. Reactome is used to classify enzymes in pathways.

Reactome is used to classify enzymes in pathways. We found 49 metabolic and catabolic pathways (*p*-value < 0.05) (Table 1).

Analysis with the IPA tool identified 14 enzymes related to tumor invasion (MME, MTHFD1, FLT4, AMFR, FER, PRKCA, ALDOB, AOX1, PSMD1, PRPS1, PRDX2, NDST1, RNASE1, F2) (Figure 8).

#### Western Blotting

Interestingly, among the enzymes that are differentially regulated, we found four enzymes—PRDX2, BLVRB, TXNL1, GAPDH, and PRDX6—that strongly up-regulate with fold change ≥ 4. PRDX2 and GADPH (Figure 9) were our primary focus because of their biologically relevant functions. An independent patient samples set of 10 controls and 10 EC was used to validate the enzymes. The Mann–Whitney sum-rank test showed a statistically significant abundance (*p* < 0.05) for PRDX2 (*p* = 0.043) while for GADPH (*p* = 0.044).

## 4. Discussion

Extracellular vesicles (EVs) are microvesicles released by cells and play significant roles in cancer. By facilitating communication between tumor cells and their microenvironment, EVs can promote cancer growth and progression such as proliferation, angiogenesis, and immune response evasion but are also involved in drug resistance. In addition, due to their presence in all body fluids, they can be exploited as non-invasive biomarkers for early cancer detection, prognosis, and monitoring treatment response [22]. Although MS has made great strides in recent years to identify thousands of proteins in EVs, the complete characterization of their proteomics remains challenging [23]. The combination of the two most common approaches, DDA and DIA, applied to shotgun proteomics, allowed the identification of biomarkers in prostate cancer [24], bladder cancer [25], ovarian cancer serum [26], and endometrial cancer [6]. In a proteomics study by Song et al. it was reported that the level of Galectin-3 binding protein (LGALS3BP), a glycoprotein found in nearly all body fluids, was elevated in plasma exosomes of EC patients and this increase was associated with the presence of metastases, which suggests this protein as a biomarker for EC [9].

In this work aiming to identify dysregulated proteins and enzymes associated with EC, we performed a comprehensive study on EVs, specifically exosomes (the best characterized EVs), by analyzing the exosome proteome using advanced MS techniques and AI tools. This research could lead to the discovery of novel biomarkers for EC, which may improve early detection and prognosis, opening avenues for non-invasive diagnostic tools, as well as shed light in new therapeutic targeting.

To allow the identification of a high number of proteins in serum exosomes, we first depleted the albumin from the serum and then precipitated exosomes with an optimized commercial kit, followed by in depth characterization. Indeed, Nanoparticle Tracking Analysis (NTA) and Western blotting marker proteins (CD63, CD9, HSC70) confirmed the exosomes’ size and concentration.

Then, we applied an integrated approach combining advanced proteomics and bioinformatics to enlarge the exosome proteome landscape. This could help the understanding of EC pathophysiology, identifying proteins and enzymes linked to tumor progression and invasion, that might offer potential targets for therapeutic interventions.

In addition to this aim, combining DIA and DDA with Chymeris (AI), we were able to identify and quantify 3628 proteins. To our knowledge, this is the study with the largest number of proteins ever identified in serum exosomes in EC. Although the acquisition of DIA greatly increases the number of proteins in our study, the use of Chymeris in the processing of DDA data further increases the protein identification allowing 222 more proteins to our list.

Looking ahead, the future of EV research holds great promise. Continued advancements in isolation and characterization techniques will enhance our ability to analyze EVs in various biological contexts. Interestingly, our work has shown that the DDA acquisition identified 222 new proteins. This indicates that this acquisition is still useful in studying depleted serum. We recommend continuing the use of DDA in combination with DIA for proteomics in the discovery serum.

By unraveling the complexities of exosomes, we can pave the way for innovative approaches to cancer diagnosis and treatment, transforming the landscape of oncology.

In the next step, we performed a bioinformatic analysis using the gProfiler tool to understand the functional impact of differently abundant proteins between EC serum exosomes versus those from controls.

The function annotation analysis highlights pathways involving neutrophil and platelet degranulation, protein phosphorylation, and metabolic processes, which play roles in tumor microenvironment modulation.

In particular, performing gProfile analysis, we found that the dysregulated protein and enzymes identified were strongly associated with metabolic pathways and/or tumor invasion mechanisms of other cancer types [27], as described briefly below, broadening their relevance also in ECs.

Neutrophils are one of the most abundant cells in the immune system involved in cancer development.

Physiologically, neutrophil degranulation is a mechanism for capturing and eliminating pathogenic micro-organisms [28], however Tumor Associated Neutrophil (TANs) are a special class of neutrophils recruited and differentiated into the tumor microenvironment via cytokines and chemokines. They are categorized into two functional types based on their activation and cytokine status: N1 TANs exhibiting antitumor activity and N2 TANs that vice versa promote tumor growth by supporting immunosuppression, angiogenesis, metastasis, and genomic instability. Interestingly, high levels of TANs are correlated with poor prognosis in cancer patients [29]. Having found an increase of neutrophil degranulation proteins in exosomes from EC should be of great interest to better characterize this protein and understand the role neutrophil in EC biology.

Similarly, platelet degranulation is another interesting pathway. Platelets play a critical role in thrombosis and hemostasis, but an important role in tumor progression and metastasis has also been described. Indeed, platelets promote tumor cell dissemination by activating endothelial cells. Vice versa, cancer cells influence platelet activation by inducing formation of platelet–tumor aggregates and triggering platelet degranulation [30,31]. Stone RL et al. propose a mechanism in which cancer cells induce the formation of platelets by the release of IL-6 [32]. In solid cancer patients, thrombosis and thrombocytopenia are common events, and a high platelet count is related to decreased survival of patients [33]. Although thrombocytopenia events are infrequent in EC [34], by our deep proteomics analysis, we identified proteins involved in the platelet degranulation process. The role of these proteins required further investigation to have a better elucidation and to shed light on EC biology. The IPA analysis showed that proteins/enzymes are related to tumor invasion in both cases. This shows that many proteins in exosomes promote tumor development.

Platelets and neutrophils have been shown to dynamically interact with tumors to promote cancer at an early stage by both cell proliferation, immune suppression, and angiogenesis, as well as metastasis and chemoresistance [35]; in this work, we suggest their involvement also in EC development. Focusing on pointed metabolic processes, the altered proteome relies on antioxidant and catalytic activity, both processes involved in tumor development [26] as also suggested by the Reactome tool annotation.

PRDX2 plays a key role in cell protection against oxidative stress by detoxifying the cell from high levels of hydrogen peroxide [36]. In addition, this enzyme can induce apoptosis in colon cancer cells by regulating oxidation. However, its overexpression is linked to the development of several tumors such as the cervix, colon, prostate, and breast [37].

BLVRB is a NADPH-dependent reductase, considered to be a modulator of cellular redox. In addition, Geiger T et al. in their study associate the enzyme with tumor progression in breast cancer cell lines, while its presence in the limbic system is associated with metastases in the same tumor [38].

TXNL1 is another enzyme involved in the oxide-reduction regulation processes protecting cells from oxidative stress damage [39].

W Xu et al. have shown that the enzyme is over-expressed in gastric cancer cells resistant to cisplatin and contributes to drug resistance [40].

PRDX6 is an enzyme localized in the cytosol that reduces H_2_O_2_, fatty acid, and phospholipid hydroperoxides [41].

In addition, this protein is overexpressed in many tumors such as the colon adenocarcinoma [42], CCA [43], and BLCA [44]. PRDX6 is overexpressed in lung adenocarcinoma and especially in patients with resistance to chemotherapy. Research results suggest that PDRX6 is considered a predictive biomarker of poor response to chemotherapy [45].

Interestingly, ROS have a dual role in cancer at early stage. They promote prolifer-ation and genomic instability sustaining tumorigenesis, and they also activate various antioxidant genes within cancer cells as a self-defense mechanism [46].

EC is often associated with metabolic syndrome, including diabetes, hypertension, and obesity. Unlike healthy cells that use efficient metabolic pathways for energy production, cancer cells undergo profound metabolic rewiring and use alternative pathways for the production of the energy they need to survive [47].

Altogether, the analysis of the exosome proteome from type I EC suggests that endometrial cancer cells to grow are exploiting all such strategies (including the cross-talk with platelets and NAT, cancer metabolism rewiring, and antioxidant response) to evade cellular and immune checkpoints.

EVs profiles can vary significantly across different cancer types, potentially offering insights into tumor biology. By comparing the composition and cargo of exosomes from EC patients with controls, we aimed to identify unique markers that may serve as diagnostic or prognostic indicators, as well as pathways that might unveil new processes involved in EC biology. To this aim, performing proteomic analysis of circulating specific exosomes, our study identified more than 40 metabolic and catabolic pathways altered in EC. This demonstrates the importance of an in-depth analysis of the exosomes’ proteome, allowing a better understanding of EC metabolism.

These proteins can serve as potential biomarkers for early detection, prognosis, and monitoring of treatment efficacy. While the current understanding of EVs in endometrial cancer is promising, further multiomics approaches that integrate proteomics, transcriptomics, and metabolomics could provide a more comprehensive view of EV biology and its impact on EC. Although this study makes significant progress in characterizing exosome proteomes in EC, its limitations highlight the need for further validation, a larger patient cohort, and functional studies to translate the results into clinical applications. In addition, lacking a longitudinal study, time course data would need to be investigated to determine whether the identified proteins are early markers or are related to progression.

Factors such as obesity, age, and hormonal status are not taken into account, which could influence the results. Addressing these issues in future research will improve the reliability and utility of the identified biomarkers.

## 5. Conclusions

In conclusion, extracellular vesicles represent a key player in endometrial cancer, with significant implications for diagnosis, prognosis, and therapy. As research continues to unravel the complexities of EVs, we may uncover new strategies for early detection and targeted treatment, ultimately improving patient outcomes in endometrial cancer. The integration of EV research into clinical practice holds the potential to transform our approach to this malignancy. Our in-depth proteomics analysis has allowed the identification of many proteins in serum exosomes. For the first time, we identified the largest number of proteins in serum exosomes in EC. Our study correlates platelet degranulation with CE and the possible risk of thrombocytopenia. These proteins are related to pathways that are important for tumor growth. In addition, the identified enzymes are related to metabolic pathways linked to tumor invasion. In our opinion, this is very important in understanding the metabolism of EC and especially in developing new therapies.

## Figures and Tables

**Figure 1 biomedicines-13-00095-f001:**
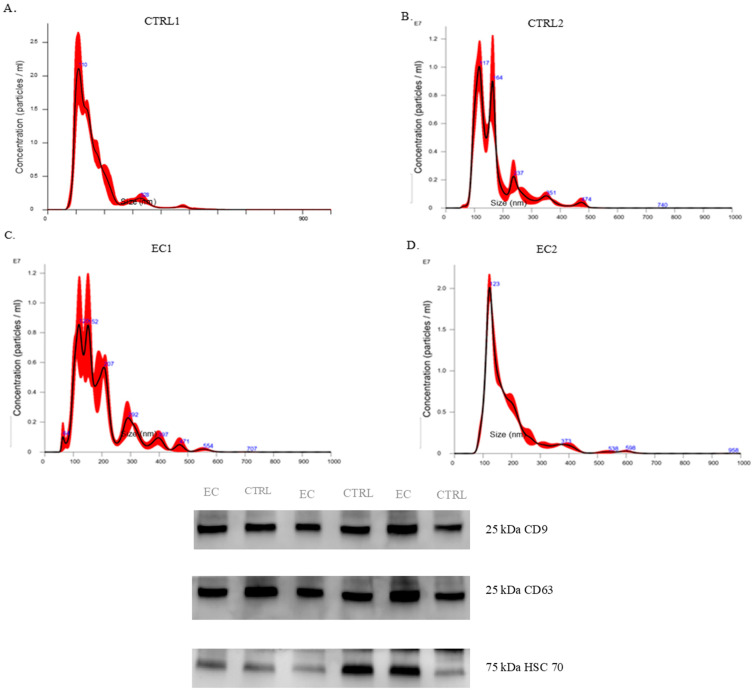
Patients and control-derived exosomes characterization. Nanoparticle concentration and size distribution of CTRL and EC, exosomes obtained through NTA. Western blot analysis of vesicle markers (CD63, CD9, HSC70). EC—endometrial cancer, Exosomes—extracellular vesicles, NTA—Nanoparticle Tracking Analysis, CTRL—control samples.

**Figure 2 biomedicines-13-00095-f002:**
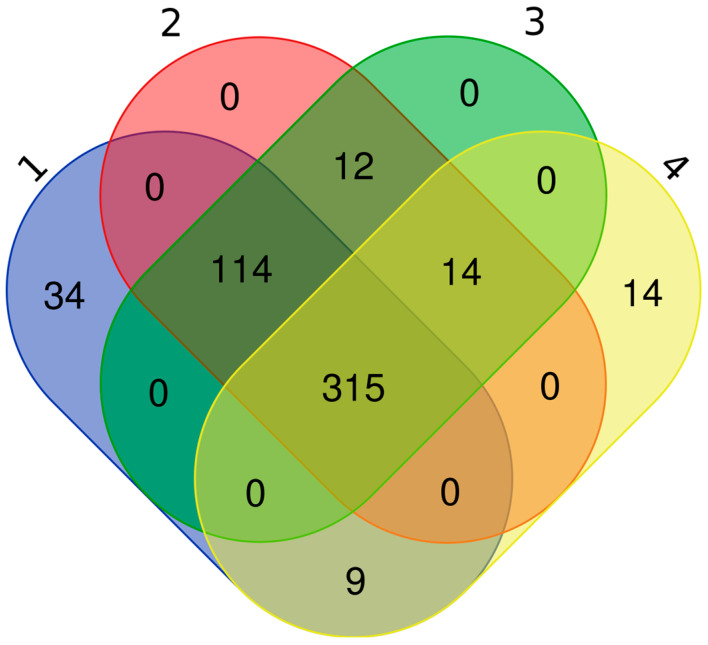
Intersection of identified proteins by four different tools: Chimery, Sequest, Amanda, and Inferys by Venn Diagram. These tools have been combined (Chimerys + Amanda and Chimerys + Amanda + Sequest, Amanda + Sequest + Inferis) to identify as many proteins as possible. 1—Chymeris, 2—Chymeris + Amanda, 3—Chymeris + Amanda + Sequest, 4—Amanda + Sequest + Inferis.

**Figure 3 biomedicines-13-00095-f003:**
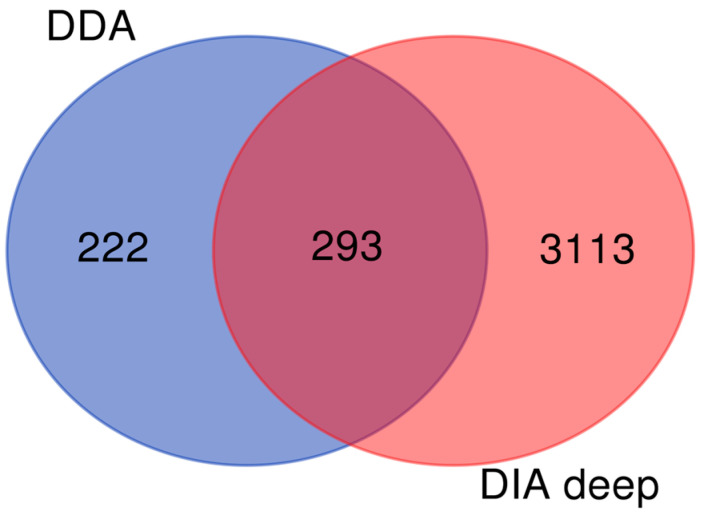
Intersection of DDA with DIA data by Venn Diagram for full proteomics data integration. DIA—data independence analysis, DDA—data dependence analysis.

**Figure 4 biomedicines-13-00095-f004:**
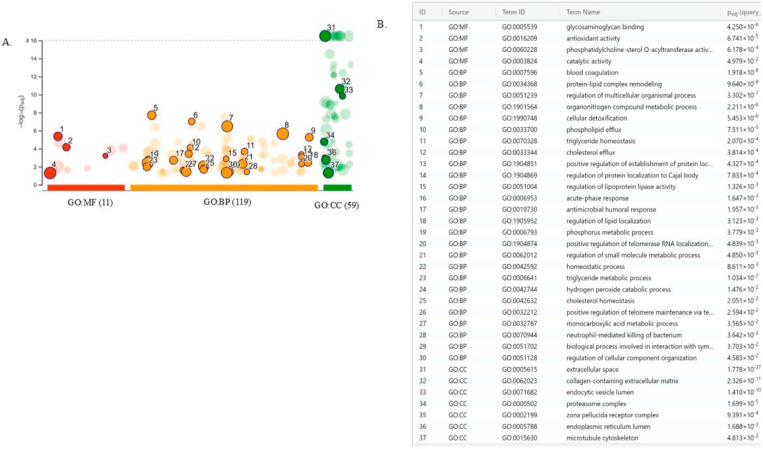
(**A**) The Manhattan plot shows the enrichment analysis results of 373 proteins and the (**B**) Denominations list of the selected circles.

**Figure 5 biomedicines-13-00095-f005:**
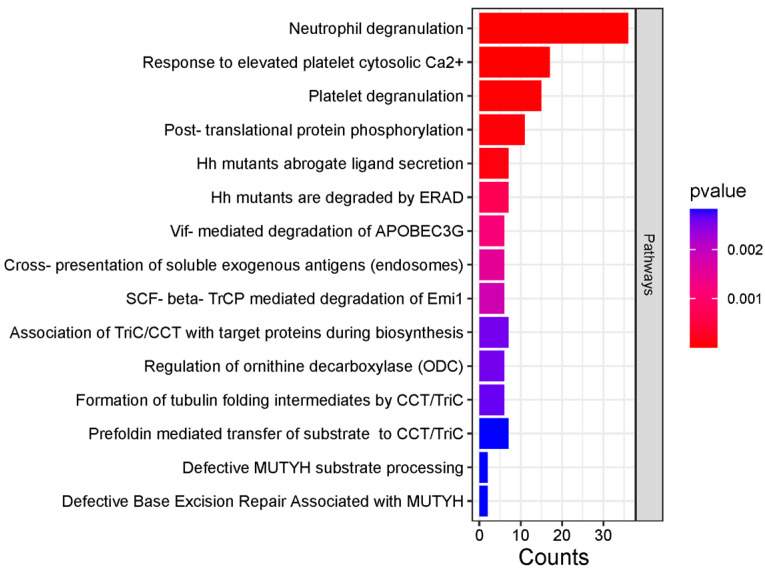
Reactome pathways were identified by 373 differentially abundant proteins. Pathways are listed according to their *p*-values and gene counts.

**Figure 6 biomedicines-13-00095-f006:**
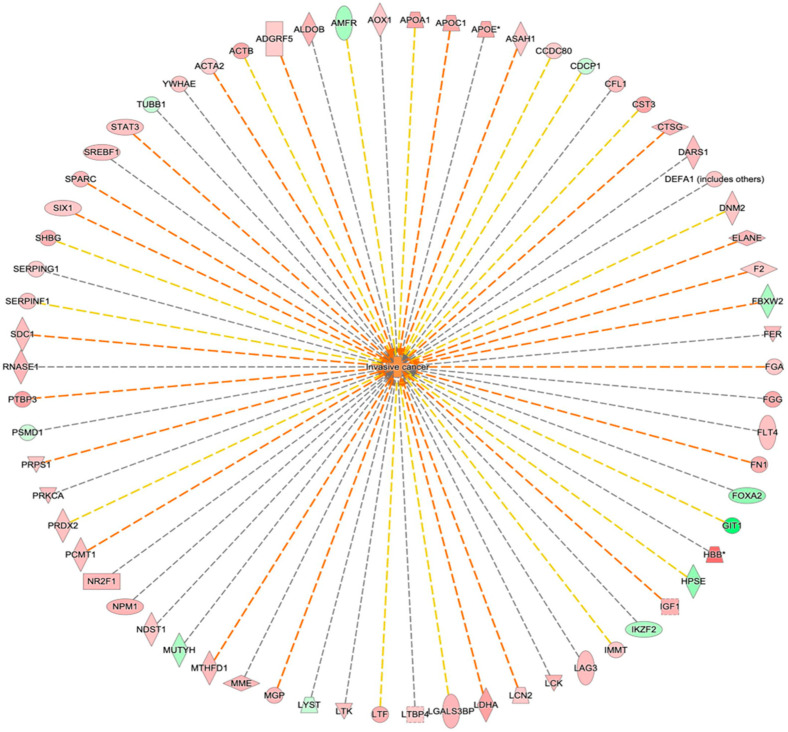
Network build-up from one of the most significant bio-functions: Invasive cancers. * Gene indicate that multiple identifiers in the dataset file map to a single gene in the Global Molecular Network.

**Figure 7 biomedicines-13-00095-f007:**
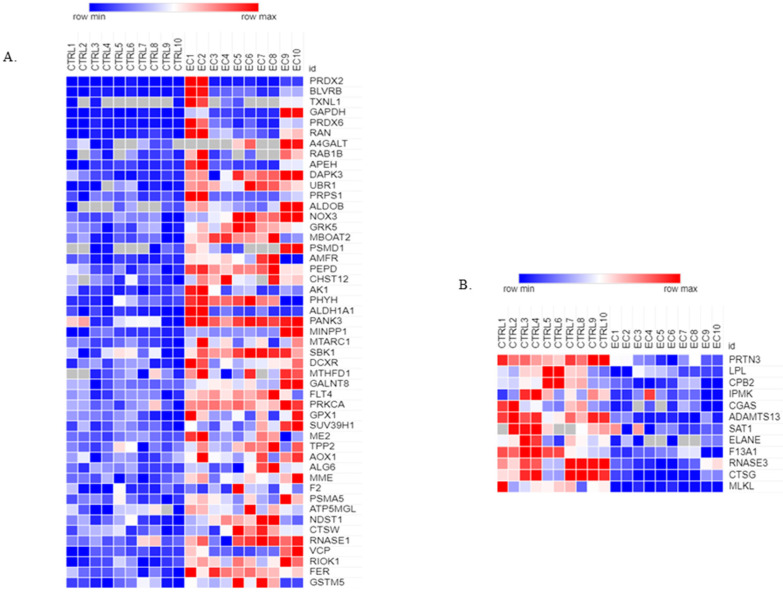
Heat map of (**A**) 49 more abundant proteins in EC; (**B**) 12 less abundant proteins in EC raw data expression proteins using the Morpheus tool.

**Figure 8 biomedicines-13-00095-f008:**
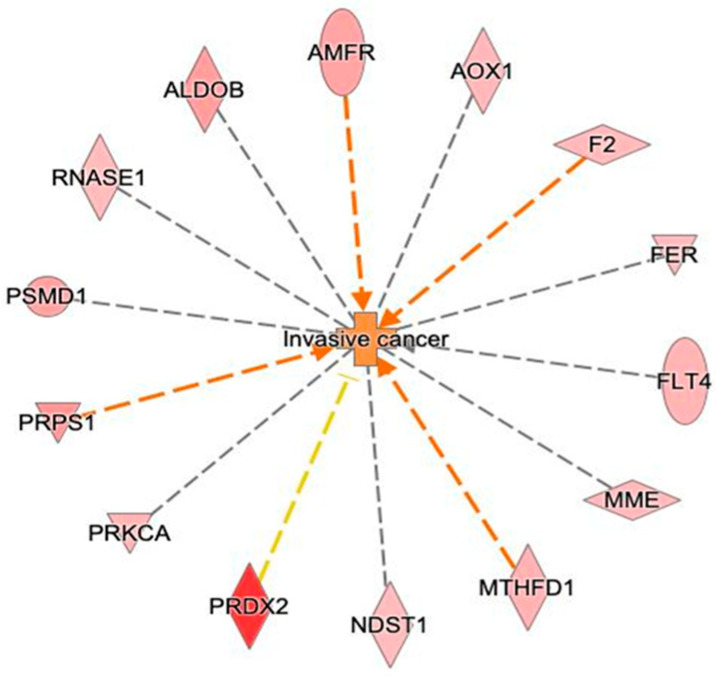
Network build-up using correlation of enzymes with invasive cancer bio-functions.

**Figure 9 biomedicines-13-00095-f009:**
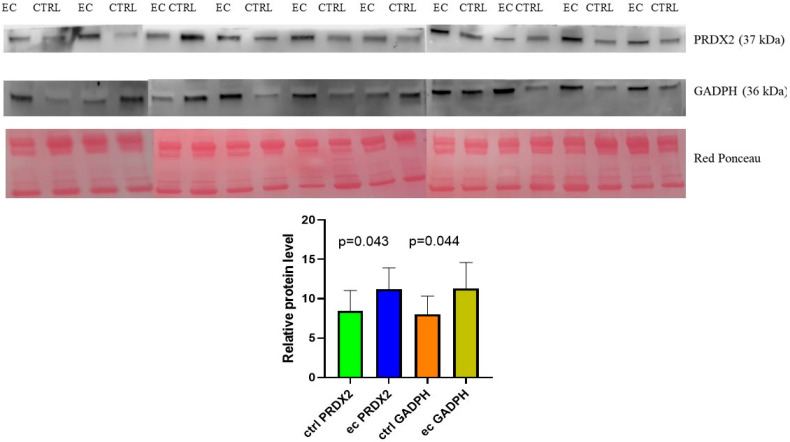
The PRDX2 and GADPH protein alteration in endometrial cancer (ec) and normal endometrium (ctrl) was confirmed using Western blot analysis. The intensity of immunostained bands was normalized against the total protein intensities measured from the same blot stained with Red Ponceau. Results are displayed as a histogram (*p* < 0.05), and each bar represents mean ± standard deviation.

**Table 1 biomedicines-13-00095-t001:** Metabolic and catabolic pathways identified by Reactome in EC serum depleted exosomes.

Pathway Identifier	Pathway Name
R-HSA-2022377	Metabolism of Angiotensinogen to Angiotensins
R-HSA-71387	Metabolism of carbohydrates
R-HSA-351202	Metabolism of polyamines
R-HSA-70263	Gluconeogenesis
R-HSA-70350	Fructose catabolism
R-HSA-5652084	Fructose metabolism
R-HSA-5663084	Diseases of carbohydrate metabolism
R-HSA-70171	Glycolysis
R-HSA-9636667	Manipulation of host energy metabolism
R-HSA-5668914	Diseases of metabolism
R-HSA-70326	Glucose metabolism
R-HSA-1483249	Inositol phosphate metabolism
R-HSA-196849	Metabolism of water-soluble vitamins and cofactors
R-HSA-1630316	Glycosaminoglycan metabolism
R-HSA-200425	Carnitine metabolism
R-HSA-196854	Metabolism of vitamins and cofactors
R-HSA-6806664	Metabolism of vitamin K
R-HSA-196757	Metabolism of folate and pterines
R-HSA-70921	Histidine catabolism
R-HSA-199220	Vitamin B5 (pantothenate) metabolism
R-HSA-9854907	Regulation of MITF-M dependent genes involved in metabolism
R-HSA-74259	Purine catabolism
R-HSA-8978934	Metabolism of cofactors
R-HSA-1793185	Chondroitin sulfate/dermatan sulfate metabolism
R-HSA-1638091	Heparan sulfate/heparin (HS-GAG) metabolism
R-HSA-189445	Metabolism of porphyrins
R-HSA-2980736	Peptide hormone metabolism
R-HSA-8956319	Nucleotide catabolism
R-HSA-70268	Pyruvate metabolism
R-HSA-975634	Retinoid metabolism and transport
R-HSA-196791	Vitamin D (calciferol) metabolism
R-HSA-1614635	Sulfur amino acid metabolism
R-HSA-194441	Metabolism of non-coding RNA
R-HSA-2142753	Arachidonic acid metabolism
R-HSA-390918	Peroxisomal lipid metabolism
R-HSA-1660662	Glycosphingolipid metabolism
R-HSA-6806667	Metabolism of fat-soluble vitamins
R-HSA-71291	Metabolism of amino acids and derivatives
R-HSA-15869	Metabolism of nucleotides
R-HSA-194068	Bile acid and bile salt metabolism
R-HSA-163685	Integration of energy metabolism
R-HSA-8978868	Fatty acid metabolism
R-HSA-428157	Sphingolipid metabolism
R-HSA-400206	Regulation of lipid metabolism by PPARalpha
R-HSA-1483257	Phospholipid metabolism
R-HSA-392499	Metabolism of proteins
R-HSA-8953854	Metabolism of RNA
R-HSA-8957322	Metabolism of steroids
R-HSA-556833	Metabolism of lipids

## Data Availability

The data presented in this study are available on request from the corresponding author. The data are not publicly available due to ethical reasons.

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
