# Peer review of "A Pilot Study of Exosome Proteomic Profiling Reveals Dysregulated Metabolic Pathways in Endometrial Cancer"

_biomedicines, 2025, doi:10.3390/biomedicines13010095_

Round 1

Reviewer 1 Report

Comments and Suggestions for Authors

The article touches on a very interesting issue, which is the proteomic analysis of enzymes of the metabolic pathway in patients with endometrial cancer. In my opinion, the manuscript needs some improvements. My minor concerns are below:

1.      Line 42 – If possible, please proved the latest data.

2.      Line 93 – Please correct the sentence: “of serum of crude serum…”. What type of serum was used?

3.      Line 97 – What information does the number 447836 refer to in this paragraph?

4.      Line 170 – In Figure 1, the diagrams are difficult to read. If possible, please improve their quality.

5.      Lines 188-195 – The fragment in question stands out from the background of the entire article, with a much larger and different font. Please, standardize the text.

6.      Line 201 – It seems to me that the article does not have 3406 tables, I think there is an error.

7.     The article did contain some minor editorial errors (e.g. sometimes it’s EV and sometimes it’s EVs; line 173, 209 where the bracket is missing; in Table 1, there is overlapping text in the first column; etc). The changes introduced would improve the readability of the manuscript.

Author Response

Line 42 – If possible, please proved the latest data.

Our reply: We included the most up-to-date information on EC incidence.

Line 93 – Please correct the sentence: “of serum of crude serum…”. What type of serum was used?

Our reply: We fixed it. We used crude serum

Line 97 – What information does the number 447836 refer to in this paragraph?

Our reply: We removed the number. It was a typing error.

Line 170 – In Figure 1, the diagrams are difficult to read. If possible, please improve their quality.

Our reply: We improved the quality of the diagrams in Figure 1.

Lines 188-195 – The fragment in question stands out from the background of the entire article, with a much larger and different font. Please, standardize the text.

Our reply: We standardized the text.

Line 201 – It seems to me that the article does not have 3406 tables, I think there is an error.

Our reply: We fixed the error.

The article did contain some minor editorial errors (e.g. sometimes it’s EV and sometimes it’s EVs; line 173, 209 where the bracket is missing; in Table 1, there is overlapping text in the first column; etc). The changes introduced would improve the readability of the manuscript.

Our reply: We fixed the errors.

Reviewer 2 Report

Comments and Suggestions for Authors

Comments and Suggestions for Authors

The authors have conducted an insightful study focusing on the proteomic signatures of extracellular vesicles (EVs) associated with endometrial cancer. While the findings are intriguing, the novelty, scientific and practical significance of findings warrant robust discussion and further investigation to confirm their reliability. Additionally, it is crucial for the authors to specify and validate marker proteins, as this will enhance the credibility and applicability of their research findings. By addressing these aspects, the authors can fortify the scientific integrity and impact of their study on this pivotal area of women's health.

Here are the suggested improvements:

Major Points

1.      Scientific and practical significance:

1.1 The current introduction lacks sufficient context to establish the foundation for the research. It is imperative to distinctly articulate the novelty and significance of this study. Numerous studies have already explored the proteome of extracellular vesicle proteins in endometrial cancer, such as those referenced by doi: 10.1038/s41388-024-03182-2 and doi: 10.1021/acs.jproteome.8b00750/.

1.2 The manuscript interchangeably uses "extracellular vesicles" and "exosomes." Considering that exosomes are a subset of extracellular vesicles, it is crucial to employ precise terminology. Furthermore, the authors should verify the classification of the isolated particles as specific types of extracellular vesicles (e.g., apoptotic bodies, microvesicles, or exosomes), as highlighted in reviews like that of Di Bella MA, 2022.

2.      Abstract:

2.1 The abstract should clearly define the study's objectives and include quantitative data and prominent results. This clarity will enhance the communication of the study's findings.

2.2 The pilot nature of the research should be acknowledged in both the title and the abstract.

3. Results section:

Given that this study performs a semi-quantitative proteomic analysis on a limited sample size, reproducibility across different experiments and settings is a concern. The authors are urged to validate their leading candidates through targeted analysis methods such as multiple reaction monitoring (MRM) or ELISA-based assays. It would also benefit the study to validate findings with independent patient samples.

3.      Discussion section:

3.1 The discussion should extract and evaluate key results, integrating recent literature. Emphasis should be placed on future research directions based on the current findings. It is essential to establish the link between protein expression changes in cancer cell lines and their corresponding levels in extracellular vesicles.

3.2 Although the study focuses on serum proteome from non-metastatic patients (Stage I), the discussion touches upon the role of EVs in tumor metastasis. These considerations need clarification in the context of early-stage cancer research.

3.3 Include a detailed exploration of the study's limitations.

4. Materials and methods:

4.1 Provide detailed inclusion and exclusion criteria for the main and control groups. Describe the method for confirming the absence of leiomyoma and adenomyosis in controls, and specify when blood samples were taken.

4.2 There is a significant age disparity between the patient groups: the median age of patients is 71 years, whereas controls are primarily of reproductive age, with a median age of 33. Given that "Endometrial cancer (EC) is the second most frequent gynecological malignancy in postmenopausal women," it is important for the control group to also comprise menopausal women. Consideration should be given to whether observed differences are due to hormonal or aging variations rather than the tumor process.

4.3 Validate the appropriateness of the sample size (10 patients in each group) through a statistical power analysis to support the findings.

Minor Points

5.      Move Figure 4 to the Supplementary.

6.      Ensure data transparency by depositing raw proteomic data in accessible repositories and including semi-quantitative data as a supplementary Excel file.

These enhancements will provide clearer insights into the findings and better align the manuscript with academic standards.

Author Response

Reviewer 2

The current introduction lacks sufficient context to establish the foundation for the research. It is imperative to distinctly articulate the novelty and significance of this study. Numerous studies have already explored the proteome of extracellular vesicle proteins in endometrial cancer, such as those referenced by doi: 10.1038/s41388-024-03182-2 and doi: 10.1021/acs.jproteome.8b00750/.

Our reply: This study aims to identify dysregulated proteins associated with EC by analyzing the proteome of exosomes with advanced MS techniques and AI tools. Although the proteome of extracellular vesicle proteins in endometrial cancer has been explored by us  (Eduardo Sommella, Valeria Capaci, Michelangelo Aloisio, Emanuela Salviati, Pietro Campiglia, Giuseppe Molinario, Danilo Licastro, Giovanni Di Lorenzo, Federico Romano, Giuseppe Ricci, Lorenzo Monasta , Blendi Ura.A Label-Free Proteomic Approach for the Identification of Biomarkers in the Exosome of Endometrial Cancer Serum. Cancers 2022, 14(24), 6262; https://doi.org/10.3390/cancers14246262 and others (Javier Mariscal, Patricia Fernandez-Puente, Valentina Calamia, Alicia Abalo, Maria Santacana, Xavier Matias-Guiu, Rafael Lopez-Lopez, Antonio Gil-Moreno, Lorena Alonso-Alconada , Miguel Abal. Proteomic Characterization of Epithelial-Like Extracellular Vesicles in Advanced Endometrial Cancer.  J Proteome Res. 2019 Mar 1;18(3):1043-1053. doi: 10.1021/acs.jproteome.8b00750), most of the previous work are in cell lines or focuses on specific proteins. In this work, we integrate cutting-edge techniques (e.g., AI-enhanced MS) to advance the understanding of EC’s molecular and metabolic basis. With this approach, we were able to broaden the landscape of exosomal proteins altered in EC patients and, with more proteins, perform more reliable pathway analysis. We agree that the rationale was lacking in the introduction and we modified it according to reviewer suggestion.

The manuscript interchangeably uses "extracellular vesicles" and "exosomes." Considering that exosomes are a subset of extracellular vesicles, it is crucial to employ precise terminology. Furthermore, the authors should verify the classification of the isolated particles as specific types of extracellular vesicles (e.g., apoptotic bodies, microvesicles, or exosomes), as highlighted in reviews like that of Di Bella MA, 2022.

Our reply: We checked this point, and according to the manufacturer's document the purification kits that we exploited are highly specific for exosomes, with insignificant amounts of some other microvesicles. The proper definition has been reported in the new version of the manuscript.

The abstract should clearly define the study's objectives and include quantitative data and prominent results. This clarity will enhance the communication of the study's findings.

Our reply: We modified the abstract explaining the objective of the study.

The pilot nature of the research should be acknowledged in both the title and the abstract.

Our reply: We changed the title and abstract following reviewer suggestion. The new title is: “A Pilot Study of exosome proteomic profiling reveals dysregulated metabolic pathways in endometrial cancer”

Given that this study performs a semi-quantitative proteomic analysis on a limited sample size, reproducibility across different experiments and settings is a concern. The authors are urged to validate their leading candidates through targeted analysis methods such as multiple reaction monitoring (MRM) or ELISA-based assays. It would also benefit the study to validate findings with independent patient samples.

Our reply: As suggested by the reviewer, we have chosen western blotting for proteomics data validation. Unfortunately, we do not have access to MRM, so we chose western blotting as an easy-to-reach target method. We validate our findings with independent patient samples. This is reported in the main text.

The discussion should extract and evaluate key results, integrating recent literature. Emphasis should be placed on future research directions based on the current findings. It is essential to establish the link between protein expression changes in cancer cell lines and their corresponding levels in extracellular vesicles.

Our reply: The discussion has been deeply handled to address the suggested points.

Although the study focuses on serum proteome from non-metastatic patients (Stage I), the discussion touches upon the role of EVs in tumor metastasis. These considerations need clarification in the context of early-stage cancer research.

Our reply: This point has been clarified during the discussion.

Include a detailed exploration of the study's limitations.

Our study: We agree that our study has some limitations, as reporter below. Although this study makes significant progress in characterizing EV proteomes in EC, its limitations highlight the need for further validation, a larger patient cohort, and functional studies to translate the results into clinical applications. In addition, lacking a longitudinal study, time course data would need to be investigated to determine whether the identified proteins are early markers or are related to progression. Factors such as obesity, age, and hormonal status are not taken into account, which could influence the results. Addressing these issues in future research will improve the reliability and utility of the identified biomarkers. These considerations were included in the  manuscript.

Provide detailed inclusion and exclusion criteria for the main and control groups. Describe the method for confirming the absence of leiomyoma and adenomyosis in controls, and specify when blood samples were taken.

Our reply: A physical examination and ultrasound were performed to screen patients for gynecological pathologies. Blood samples were taken after gynecological examination and before the surgery. The inclusion criteria include Patients over 18 years of age for both groups. Patients with endometrial carcinoma (endometrioid adenocarcinoma) with post-hysteroscopic histological diagnosis. (stage FIGO I) for the patients’ group. Patients with non-endometrial pathology with negative endometrial histology as a control group. The exclusion criteria for both groups include patients with infectious diseases (HIV, HBV, HCV, etc.), leiomyoma, and adenomyosis. Presence of synchronous tumors. Patients suffering from other neoplastic pathologies and/or being subjected to chemo and radiotherapy in the last 10 years. We have included the answer in the text

There is a significant age disparity between the patient groups: the median age of patients is 71 years, whereas controls are primarily of reproductive age, with a median age of 33. Given that "Endometrial cancer (EC) is the second most frequent gynecological malignancy in postmenopausal women," it is important for the control group to also comprise menopausal women. Consideration should be given to whether observed differences are due to hormonal or aging variations rather than the tumor process.

Our reply: We thank the reviewer for the questions. For this study, we use younger patients as the control group to compare them with the case group. We selected patients who did not have comorbidities, didn’t take medications, and had no inflammatory issues. We made this choice to avoid the influences on serum protein abundance that could affect the proteomics analysis. We believe that creating a comparison between pathological groups and an ideal group of “perfect” patients could further strengthen our data. The comparison groups should be as similar as possible in terms of age, comorbidities, hormonal status, weight, etc. Ideally, there should be two identical groups that differ only in the pathology under study. It is not enough to consider only age; all factors must be considered. Given the difficulty in creating two identical groups that differ only in the presence of EC, we preferred to compare the cases with an ideal croup.

4.3 Validate the appropriateness of the sample size (10 patients in each group) through a statistical power analysis to support the findings.

Our reply: We used a bioinformatic tool from MD Anderson Cancer Center (https://biostatistics.mdanderson.org/MicroarraySampleSize/) for statistical power analysis. We have considered a number of genes to be tested equal to 3600, and only one false positive is acceptable. We set the required fold difference to 2, and the study power to 80%, assuming a standard deviation (SD) of the intensity measures of the genes equal to 0.7 on a base 2 logarithmic scale (an SD equal to 0.7 is considered a realistic hypothesis for genes with moderate or high expression levels). It is therefore obtained that each group should have size 10, with an alpha per gene value of 0.00028. Considering that there are 2 groups to be compared, the minimum expected number will be 20.

Minor Points

Move Figure 4 to the Supplementary.

Our reply: I moved Figure 4 in the supplementary.

Ensure data transparency by depositing raw proteomic data in accessible repositories and including semi-quantitative data as a supplementary Excel file.

Our reply: The mass spectrometry proteomics data have been deposited to the ProteomeXchange Consortium via the PRIDE partner repository with the dataset identifier PXD058193. We included semi-quantitative data as a supplementary Excel file.

Reviewer 2

The current introduction lacks sufficient context to establish the foundation for the research. It is imperative to distinctly articulate the novelty and significance of this study. Numerous studies have already explored the proteome of extracellular vesicle proteins in endometrial cancer, such as those referenced by doi: 10.1038/s41388-024-03182-2 and doi: 10.1021/acs.jproteome.8b00750/.

Our reply: This study aims to identify dysregulated proteins associated with EC by analyzing the proteome of exosomes with advanced MS techniques and AI tools. Although the proteome of extracellular vesicle proteins in endometrial cancer has been explored by us  (Eduardo Sommella, Valeria Capaci, Michelangelo Aloisio, Emanuela Salviati, Pietro Campiglia, Giuseppe Molinario, Danilo Licastro, Giovanni Di Lorenzo, Federico Romano, Giuseppe Ricci, Lorenzo Monasta , Blendi Ura.A Label-Free Proteomic Approach for the Identification of Biomarkers in the Exosome of Endometrial Cancer Serum. Cancers 2022, 14(24), 6262; https://doi.org/10.3390/cancers14246262 and others (Javier Mariscal, Patricia Fernandez-Puente, Valentina Calamia, Alicia Abalo, Maria Santacana, Xavier Matias-Guiu, Rafael Lopez-Lopez, Antonio Gil-Moreno, Lorena Alonso-Alconada , Miguel Abal. Proteomic Characterization of Epithelial-Like Extracellular Vesicles in Advanced Endometrial Cancer.  J Proteome Res. 2019 Mar 1;18(3):1043-1053. doi: 10.1021/acs.jproteome.8b00750), most of the previous work are in cell lines or focuses on specific proteins. In this work, we integrate cutting-edge techniques (e.g., AI-enhanced MS) to advance the understanding of EC’s molecular and metabolic basis. With this approach, we were able to broaden the landscape of exosomal proteins altered in EC patients and, with more proteins, perform more reliable pathway analysis. We agree that the rationale was lacking in the introduction and we modified it according to reviewer suggestion.

The manuscript interchangeably uses "extracellular vesicles" and "exosomes." Considering that exosomes are a subset of extracellular vesicles, it is crucial to employ precise terminology. Furthermore, the authors should verify the classification of the isolated particles as specific types of extracellular vesicles (e.g., apoptotic bodies, microvesicles, or exosomes), as highlighted in reviews like that of Di Bella MA, 2022.

Our reply: We checked this point, and according to the manufacturer's document the purification kits that we exploited are highly specific for exosomes, with insignificant amounts of some other microvesicles. The proper definition has been reported in the new version of the manuscript.

The abstract should clearly define the study's objectives and include quantitative data and prominent results. This clarity will enhance the communication of the study's findings.

Our reply: We modified the abstract explaining the objective of the study.

The pilot nature of the research should be acknowledged in both the title and the abstract.

Our reply: We changed the title and abstract following reviewer suggestion. The new title is: “A Pilot Study of exosome proteomic profiling reveals dysregulated metabolic pathways in endometrial cancer”

Given that this study performs a semi-quantitative proteomic analysis on a limited sample size, reproducibility across different experiments and settings is a concern. The authors are urged to validate their leading candidates through targeted analysis methods such as multiple reaction monitoring (MRM) or ELISA-based assays. It would also benefit the study to validate findings with independent patient samples.

Our reply: As suggested by the reviewer, we have chosen western blotting for proteomics data validation. Unfortunately, we do not have access to MRM, so we chose western blotting as an easy-to-reach target method. We validate our findings with independent patient samples. This is reported in the main text.

The discussion should extract and evaluate key results, integrating recent literature. Emphasis should be placed on future research directions based on the current findings. It is essential to establish the link between protein expression changes in cancer cell lines and their corresponding levels in extracellular vesicles.

Our reply: The discussion has been deeply handled to address the suggested points.

Although the study focuses on serum proteome from non-metastatic patients (Stage I), the discussion touches upon the role of EVs in tumor metastasis. These considerations need clarification in the context of early-stage cancer research.

Our reply: This point has been clarified during the discussion.

Include a detailed exploration of the study's limitations.

Our study: We agree that our study has some limitations, as reporter below. Although this study makes significant progress in characterizing EV proteomes in EC, its limitations highlight the need for further validation, a larger patient cohort, and functional studies to translate the results into clinical applications. In addition, lacking a longitudinal study, time course data would need to be investigated to determine whether the identified proteins are early markers or are related to progression. Factors such as obesity, age, and hormonal status are not taken into account, which could influence the results. Addressing these issues in future research will improve the reliability and utility of the identified biomarkers. These considerations were included in the  manuscript.

Provide detailed inclusion and exclusion criteria for the main and control groups. Describe the method for confirming the absence of leiomyoma and adenomyosis in controls, and specify when blood samples were taken.

Our reply: A physical examination and ultrasound were performed to screen patients for gynecological pathologies. Blood samples were taken after gynecological examination and before the surgery. The inclusion criteria include Patients over 18 years of age for both groups. Patients with endometrial carcinoma (endometrioid adenocarcinoma) with post-hysteroscopic histological diagnosis. (stage FIGO I) for the patients’ group. Patients with non-endometrial pathology with negative endometrial histology as a control group. The exclusion criteria for both groups include patients with infectious diseases (HIV, HBV, HCV, etc.), leiomyoma, and adenomyosis. Presence of synchronous tumors. Patients suffering from other neoplastic pathologies and/or being subjected to chemo and radiotherapy in the last 10 years. We have included the answer in the text

There is a significant age disparity between the patient groups: the median age of patients is 71 years, whereas controls are primarily of reproductive age, with a median age of 33. Given that "Endometrial cancer (EC) is the second most frequent gynecological malignancy in postmenopausal women," it is important for the control group to also comprise menopausal women. Consideration should be given to whether observed differences are due to hormonal or aging variations rather than the tumor process.

Our reply: We thank the reviewer for the questions. For this study, we use younger patients as the control group to compare them with the case group. We selected patients who did not have comorbidities, didn’t take medications, and had no inflammatory issues. We made this choice to avoid the influences on serum protein abundance that could affect the proteomics analysis. We believe that creating a comparison between pathological groups and an ideal group of “perfect” patients could further strengthen our data. The comparison groups should be as similar as possible in terms of age, comorbidities, hormonal status, weight, etc. Ideally, there should be two identical groups that differ only in the pathology under study. It is not enough to consider only age; all factors must be considered. Given the difficulty in creating two identical groups that differ only in the presence of EC, we preferred to compare the cases with an ideal croup.

4.3 Validate the appropriateness of the sample size (10 patients in each group) through a statistical power analysis to support the findings.

Our reply: We used a bioinformatic tool from MD Anderson Cancer Center (https://biostatistics.mdanderson.org/MicroarraySampleSize/) for statistical power analysis. We have considered a number of genes to be tested equal to 3600, and only one false positive is acceptable. We set the required fold difference to 2, and the study power to 80%, assuming a standard deviation (SD) of the intensity measures of the genes equal to 0.7 on a base 2 logarithmic scale (an SD equal to 0.7 is considered a realistic hypothesis for genes with moderate or high expression levels). It is therefore obtained that each group should have size 10, with an alpha per gene value of 0.00028. Considering that there are 2 groups to be compared, the minimum expected number will be 20.

Minor Points

Move Figure 4 to the Supplementary.

Our reply: I moved Figure 4 in the supplementary.

Ensure data transparency by depositing raw proteomic data in accessible repositories and including semi-quantitative data as a supplementary Excel file.

Our reply: The mass spectrometry proteomics data have been deposited to the ProteomeXchange Consortium via the PRIDE partner repository with the dataset identifier PXD058193. We included semi-quantitative data as a supplementary Excel file.

Round 2

Reviewer 1 Report

Comments and Suggestions for Authors

The authors have made a significant correction. The manuscript has been rewritten in accordance with the reviewer's suggestions. I accept the manuscript for publication.

Reviewer 2 Report

Comments and Suggestions for Authors

The authors took into account all the comments and made the necessary improvements to the text of the article.